# Remdesivir Derivative VV116 Is a Potential Broad-Spectrum Inhibitor of Both Human and Animal Coronaviruses

**DOI:** 10.3390/v15122295

**Published:** 2023-11-23

**Authors:** Weiyong Liu, Min Zhang, Chengxiu Hu, Huijuan Song, Yi Mei, Yingle Liu, Qi Zhang

**Affiliations:** 1Department of Laboratory Medicine, Tongji Hospital, Tongji Medical College, Huazhong University of Science and Technology, Wuhan 430030, China; wyliu@hust.edu.cn (W.L.); huchengxiuamma@163.com (C.H.); shj18134682591@163.com (H.S.); 2Department of Emergency Medicine, The Third People’s Hospital of Hubei Province, Wuhan 430033, China; zhangmin_701@163.com; 3State Key Laboratory of Virology, College of Life Sciences, Wuhan University, Wuhan 430072, China; 15972986161@163.com

**Keywords:** remdesivir, VV116, coronavirus, broad-spectrum anti-coronavirus activity

## Abstract

Coronaviruses represent a significant threat to both human and animal health, encompassing a range of pathogenic strains responsible for illnesses, from the common cold to more severe diseases. VV116 is a deuterated derivative of Remdesivir with oral bioavailability that was found to potently inhibit SARS-CoV-2. In this work, we investigated the broad-spectrum antiviral activity of VV116 against a variety of human and animal coronaviruses. We examined the inhibitory effects of VV116 on the replication of the human coronaviruses HCoV-NL63, HCoV-229E, and HCoV-OC43, as well as the animal coronaviruses MHV, FIPV, FECV, and CCoV. The findings reveal that VV116 effectively inhibits viral replication across these strains without exhibiting cytotoxicity, indicating its potential for safe therapeutic use. Based on the results of a time-of-addition assay and an rNTP competitive inhibition assay, it is speculated that the inhibitory mechanism of VV116 against HCoV-NL63 is consistent with its inhibition of SARS-CoV-2. Our work presents VV116 as a promising candidate for broad-spectrum anti-coronavirus therapy, with implications for both human and animal health, and supports the expansion of its therapeutic applications as backed by detailed experimental data.

## 1. Introduction

The coronaviruses are enveloped viruses with positive-sense single-stranded RNA genomes belonging to the family Coronaviridae. Their infections frequently cause respiratory diseases in humans. The first human coronavirus was successfully isolated as early as 1965 [1], followed by the discovery of other human coronaviruses. To date, there are seven coronaviruses with the ability to infect humans, four of which are low-pathogenic coronaviruses, including HCoV-NL63, HCoV-229E, HCoV-HKU1, and HCoV-OC43, which are responsible for 15–30% of global cases of the common cold [2]. The first two coronaviruses belong to the genus Alphacoronavirus, and the latter two belong to the genus Betacoronavirus. Usually, these four coronaviruses only cause a mild cold; however, they can cause bronchitis and pneumonia and lead to death in rare cases [3,4]. The remaining three human coronaviruses, including SARS-CoV, MERS-CoV, and SARS-CoV-2, all of which belong to the genus Betacoronavirus, are highly pathogenic. The mortality of SARS-CoV and MERS-CoV is 10% and 34.56%, respectively [5,6]. SARS-CoV-2, whose initial mortality was around 3.4% early on in the epidemic, has become less virulent but more transmissible and still presents a major threat to human health [7]. In addition to humans, coronaviruses also have the ability to infect different animals, including farm animals (pigs, horses, and cattle), companion animals (cats and dogs), laboratory animals (rats and mice), and wild animals. Four animal coronaviruses were considered in this investigation: mouse hepatitis virus (MHV), feline infectious peritonitis virus (FIPV), feline enteric coronavirus (FECV), and canine coronavirus (CCoV). Of these, mouse hepatitis virus is the most extensively studied. Animals infected with this virus have been used as models for many virological and clinical studies [8]. Feline coronavirus was first identified in 1970 via electron microscopy [9]. There are two virulence variants of feline coronavirus in domestic and wild cats that cause severe enteritis or infectious peritonitis: feline enteric coronavirus (FECV) and feline infectious peritonitis virus (FIPV). It is believed that FIPV is the result of a mutation of FECV that occurred during viral replication [10]. Canine coronavirus infection was first reported in 1971 [11]. Typically, the infection causes mild or asymptomatic forms of enteritis in the gastrointestinal tract [12].

Nucleoside analogs are an important type of antiviral drug that target the conserved activity site of viral polymerase and integrate into the newly synthesized genome product of the virus in the form of nucleotides, leading to chain termination [13,14]. When confronting the SARS-CoV-2 epidemic, nucleoside analogs were the first candidate drug type to be proposed. Remdesivir (Figure 1A) was the first anti-SARS-CoV-2 nucleoside analog approved by the FDA [15]. It was originally developed by Gilead Sciences for the treatment of Ebola [16] and was later found to inhibit the replication of a wide range of coronaviruses, including SARS-CoV and MERS-CoV [17,18,19]. It is a prodrug of a nucleoside analog that transforms into GS-441524 monophosphate (Figure 1A) intracellularly and finally into GS-441524 triphosphate, an analog of adenosine nucleoside triphosphate, influencing the polymerase activity of SARS-CoV-2 and inhibiting the elongation of the RNA product strand [20]. The EC_50_ of Remdesivir against SARS-CoV-2 was found to be 0.77~4.9 μM [21,22,23,24]. Recently, a Remdesivir derivative named VV116 (Figure 1A) has been successfully developed, showing excellent anti-SARS-CoV-2 ability in vitro and in vivo [24,25,26]. Furthermore, VV116 has good oral bioavailability, which effectively negates the need for targeting Remdesivir to the liver via intravenous injection, which affects its therapeutic effect on pulmonary viral infections [24,27]. VV116 has exhibited satisfactory safety, tolerability, and pharmacokinetic properties in phase I clinical trials [28]. In a phase III clinical trial (NCT05341609), it was found that a 5-day course of oral treatment with VV116 was noninferior to Nirmatrelvir–Ritonavir treatment in shortening the time to sustained clinical recovery [26]. There is currently another completed phase III clinical trial (NCT05582629) awaiting publication, and we believe that the findings will be disclosed shortly. In a prospective cohort study conducted in Chinese participants infected with SARS-CoV-2 omicron variants, VV116 was found to reduce the SARS-CoV-2 nucleic acid shedding time by 2–3 days (8.56 days in the VV116 group vs. 11.13 days in the control group) [29]. On 28 January 2023, VV116 was granted accelerated approval by the Chinese National Medical Products Administration (NMPA) for the treatment of adult patients with mild-to-moderate COVID-19 [30].

In this study, the broad-spectrum anti-coronavirus activity of VV116 was confirmed with regard to both human coronaviruses (HCoV-NL63, HCoV-229E, and HCoV-OC43) and animal coronaviruses (MHV, FIPV, FECV, and CCoV). With the results of this study, we provide substantial support for a comprehensive understanding of the ability of VV116 to fight against coronaviruses.

## 2. Materials and Methods

### 2.1. Cell Lines, Viruses, and Reagents

The cell lines were obtained from the China Center for Type Culture Collection (CCTCC). Human fibroblast cell line MRC-5 (GDC0032), human colorectal adenocarcinoma cell line Caco-2 (GDC0153), feline kidney cell line CRFK (GDC0321), and murine fibroblast cell line NCTC clone 929 (GDC0034) were maintained in Eagle’s minimum essential medium (Gibco, Billings, MT, USA). Human epithelial cell line HCT-8 (GDC0201) was maintained in RPMI-1640 medium (Gibco). The media for MRC-5, CRFK, and NCTC clone 929 were supplemented with fetal bovine serum (Gibco). The medium for Caco-2 was supplemented with 20% fetal bovine serum (Gibco). The medium for HCT-8 was supplemented with 10% horse serum (HyClone, Logan, UT, USA).

The viruses were obtained from the China Center for Type Culture Collection (CCTCC) or previously isolated from clinical samples. HCoV-NL63 was propagated and titrated with the Caco-2 cell line; HCoV-OC43 was propagated and titrated with the HCT-8 cell line; HCoV-229E was propagated and titrated with the MRC-5 cell line; and mouse hepatitis virus (MHV) was propagated and titrated with the NCTC clone 929 cell line. Feline infectious peritonitis virus (FIPV), feline enteric coronavirus (FECV), and canine coronavirus (CCoV) were propagated and titrated with the CRFK cell line. The nucleoside analogs Remdesivir and heparan sulfate were purchased from Topscience (Shandong, China). The deuremidevir hydrobromide tablets (VV116) were purchased from Junshi Biosciences (Beijing, China). The antibodies used in this study included HCoV-NL63 nucleoprotein antibody (40641-T62, SinoBiological, Shenzhen, China) and GAPDH antibody (60004-1-Ig, Proteintech, Wuhan, China).

### 2.2. Cytotoxicity Assay

The cytotoxicity of the nucleoside analog used in this study was evaluated in different cell lines with a Cell Counting Kit-8 (Vazyme, Nanjing, China). Cells were cultured in a 96-well plate with 80–90% confluence (approximately 4 × 10^4^ cells/well) and treated with various concentrations of nucleoside analog as indicated for 48 h. Then, each well of cells was treated with 10 μL of the CCK-8 solution and incubated for 2 h. Absorbance at OD450 nm was measured with a microplate reader, and the medium without cells was used as a blank control. Cell viability was determined as the percentage of that of the control. CC_50_ was calculated using a nonlinear regression model (four parameters) in Prism 9 software. The cytotoxicity assay was performed in triplicate and repeated in three independent experiments.

### 2.3. Viral Yield Reduction Assay

The antiviral ability of nucleoside analogs against human coronaviruses and animal coronaviruses was measured using the viral yield reduction (VYR) assay. Viruses were propagated in the presence of various concentrations of nucleoside analogs, as indicated. The progeny viruses in the supernatant were collected 24 h post-infection, and the viral titers were determined. EC_50_ was calculated by plotting the percentage of virus yields against various concentrations of the nucleoside analog using a dose–response curve in Prism 9 software. The viral yield reduction assay was performed in triplicate and repeated in three independent experiments.

### 2.4. RNA Extraction and Real-Time PCR

Total RNA was extracted from virus-infected cells 48 h after infection using TRIzol reagent (Thermo Fisher Scientific, Waltham, MT, USA) and reverse-transcribed using PrimeScript RT Master Mix and Random 6 mers (TAKARA, Dalian, China). Viral RNA was amplified on LightCycler 480 (Roche, Basel, Switzerland) using AceQ qPCR SYBR Green Master Mix (Vazyme, Nanjing, China) and the primers indicated in Table 1. The amplification conditions were as follows: 95 °C for 5 min, followed by 40 cycles of 95 °C for 10 s and 60 °C for 30 s. A melting curve analysis was performed to verify the specificity of each amplification. All experiments were performed in triplicate and repeated three times independently.

### 2.5. Time-of-Addition Assay

Cells were seeded in a 12-well plate at a density of 2 × 10^5^ cells/well and inoculated with HCoV-NL63 at a multiplicity of infection (MOI) of 0.1. As depicted in Figure 3A, the compounds (2 μM VV116, 2 μM Remdesivir, and 300 μg/mL heparan sulfate) were added at different stages of virus infection, as illustrated in Figure 3A. In the “full time” group, cells were pre-treated with the compounds for one hour, followed by two hours of infection in the presence of the compounds. The cells were then washed with PBS and grown in fresh medium containing the compounds until the end of the experiment. In the “Entry” group, cells were treated with the compounds for 1 h prior to infection, followed by infection for 2 h in the presence of the compounds. Then, the cells were washed with PBS and cultured in fresh medium devoid of the compounds until the end of the experiment. In the “Post-entry” group, cells were inoculated with the virus for 2 h, washed with PBS, and then grown in fresh medium containing the compounds until the end of the experiment. Cells were harvested 24 h post-infection. Western blot was used to detect the expression of viral nucleoprotein, and GAPDH was used as an internal control.

### 2.6. Plaque Assay

Cells were seeded in 24-well plates to obtain an 80% confluent cell monolayer. The next day, the viral supernatant of different samples was added to each well. After absorption for 1 h, the viruses were removed, and the infected cells were overlaid with medium containing 0.6% Avicel (Sigma, St. Louis, MO, USA). Five days after infection, cells were fixed with 4% formaldehyde and stained with 1% crystal violet. The plates were washed, and the plaques were counted.

### 2.7. Immunofluorescence Assay

Cells were seeded in a 24-well plate and infected with HCoV-NL63 at an MOI of 0.1. Various concentrations of VV116, as illustrated in Figure 2E, were added 1 h post-infection. Twenty-four hours after infection, cells were fixed with 4% paraformaldehyde and permeabilized with 0.1% Triton X-100 (Sigma, St. Louis, MO, USA). Cells were incubated with 1% bovine serum albumin for blocking and then incubated with HCoV-NL63 nucleoprotein antibody overnight at 4 °C. The next day, FITC-conjugated secondary antibody and DAPI were added for 1 h of incubation, and the result was visualized with a fluorescence microscope.

### 2.8. rNTP Competitive Inhibition Assay

Cells were seeded in a 12-well plate and infected with HCoV-NL63 at an MOI of 0.1. VV116 (2 μM) or HS (300 μg/mL) was added with different concentrations of rNTP (Promega, Madison, WI, USA), as indicated in Figure 3B. Cells were harvested 24 h post-infection. Western blot was used to detect the expression of viral nucleoprotein, and GAPDH was used as an internal control.

### 2.9. In Vivo Study

Three- to four-week-old specific-pathogen-free (SPF) BALB/c mice were maintained in a biosafety level 2 (BSL-2) facility. Mice were randomly divided into 5 groups (n = 6 for each group), including the mock group; the control group, which was treated with the vehicle (40% PEG400, 10% Kolliphor HS15, and 50% ultrapure water); and the experiment groups, which were treated with VV116 orally bis in die (25 mg/kg, 50 mg/kg, and 100 mg/kg). Mice were anesthetized via the intraperitoneal injection of 2.5% Avertin and inoculated intranasally with MHV-A59 at 1 × 10^5^ plaque-forming units (PFUs). One hour after MHV infection, mice were treated with the vehicle or VV116 according to the group description above. Mice were monitored daily for symptoms of disease, including changes in body weight and death, until 7 days after infection.

Another experiment was performed for virus titration detection. Mice were randomly divided into 4 groups (n = 6 for each group), including the vehicle-treated control group and the experiment groups treated with VV116 orally bis in die (25 mg/kg, 50 mg/kg, and 100 mg/kg). The experiment was performed following the same procedure as described above. Three days after infection, mice were sacrificed to collect liver tissues, and the virus titers present were identified.

### 2.10. Statistical Analysis

All experiments were repeated at least three times independently. All values are presented as the mean ± standard deviation (SD) of individual samples. Data analysis was performed with GraphPad Prism software. *p*-values of less than or equal to 0.05 were considered statistically significant (*, *p* ≤ 0.05. **, *p* ≤ 0.01. ***, *p* ≤ 0.001).

## 3. Results

### 3.1. VV116 Exhibits Broad-Spectrum Antiviral Activity against Human and Animal Coronaviruses

To explore the influence of VV116 on the replication of coronaviruses, we first evaluated its antiviral effects against three human coronaviruses (HCoV-NL63, HCoV-229E, and HCoV-OC43) with a viral yield reduction (VYR) assay in Caco-2, MRC-5, and HCT-8 cells, respectively. The results show that VV116 inhibited the replications of HCoV-NL63, HCoV-229E, and HCoV-OC43 with EC_50_ values of 2.097 ± 0.026 μM, 2.351 ± 0.072 μM, and 6.268 ± 0.123 μM. No obvious cytotoxicity of the compound was observed with the concentrations used in the experiment. The antiviral effect of Remdesivir, which was used as a positive control in this study, against those human coronaviruses was also evaluated. The results show that Remdesivir inhibited the replications of HCoV-NL63, HCoV-229E, and HCoV-OC43 with EC_50_ values of 0.495 ± 0.01 μM, 0.056 ± 0.002 μM, and 0.069 ± 0.001 μM, in agreement with the findings of other studies (Figure 1B). We then investigated the antiviral effects of VV116 against animal coronaviruses (mouse hepatitis virus (MHV), feline infectious peritonitis virus (FIPV), feline enteric coronavirus (FECV), and canine coronavirus (CCoV)) in NCTC clone 929 and CRFK cells. The results show that VV116 also inhibited the replications of MHV, FIPV, FECV, and CCoV with EC_50_ values of 1.498 ± 0.017 μM, 0.665 ± 0.022 μM, 0.847 ± 0.026 μM, and 0.79 ± 0.015 μM. However, no obvious cytotoxicity against NCTC clone 929 or CRFK cells was observed (Figure 1C). Overall, VV116 exhibited broad-spectrum anti-coronavirus activity.

### 3.2. Confirmation of the Antiviral Activity of VV116 against HCoV-NL63 and FIPV

Among all the human and animal coronaviruses considered in this study, HCoV-NL63 and FIPV showed the most obvious inhibitory response to VV116. This inhibitory response was further confirmed in various ways. A series of dose-dependent experiments were performed, and different indicators of viral replication were detected. Intracellular viral RNA expression gradually decreased as the concentration of VV116 increased. When the concentration of VV116 was 0.1 μM, the viral RNA expression of HCoV-NL63 exhibited a statistically significant reduction compared with the control group. When the concentration of VV116 reached 1 μM, the viral RNA expression of the experimental group was half that of the control group. When the concentration of VV116 exceeded 10 μM, the viral RNA expression of the experimental group was less than a quarter of that of the control group (Figure 2A, left panel). FIPV was even more sensitive to the inhibitory effect of VV116. Its intracellular viral RNA expression dropped significantly upon the introduction of VV116 at a concentration of 0.01 μM. When the concentration of VV116 reached 1 μM, its viral RNA expression was half that of the control group. Finally, its viral RNA was barely detectable at a concentration above 100 μM (Figure 2A, right panel).

A portion of the cell samples in the dose-dependent experiment were subjected to viral protein expression detection. The overall trend of the result is consistent with that at the RNA level. The nucleoprotein expression of HCoV-NL63 was half that of the control group upon the introduction of VV116 at a concentration of 1 μM, and it was barely visible when the concentration reached 10 μM (Figure 2B). The protein expression of FIPV was not evaluated because no commercial antibody for this was available.

To further confirm the antiviral activity of VV116, the viral titers in the samples for the dose-dependent experiment were measured. The results of the 50% tissue culture infectious dose (TCID_50_) assay are shown in Figure 2C. The viral titer of HCoV-NL63 dropped to about half that of the control group when the concentration of VV116 reached 1 μM, and the viral titer was less than a quarter of that of the control group when the concentration of VV116 exceeded 10 μM. The results for FIPV are similar to those for HCoV-NL63, revealing that VV116 has substantial anti-coronavirus activity. In order to make the antiviral activity of VV116 more intuitive, a plaque assay (Figure 2D) and immunofluorescence experiment (Figure 2E) were performed. The overall trend of the results was the same. In general, VV116 exhibited substantial anti-coronavirus activity, coupled with low cytotoxicity, making it a promising compound worthy of further research.

### 3.3. Mechanism of Antiviral Activity

HCoV-NL63 was used to investigate the mechanism of VV116′s antiviral activity. A time-of-addition assay was used to determine at which step in the viral life cycle this compound is involved. Specific concentrations of VV116, Remdesivir, and heparan sulfate were added to the cell supernatant at different time points, as indicated in Figure 3A. Twenty-four hours post-infection, samples of each treatment group were collected and subjected to a Western blot for the detection of viral nucleoprotein expression. VV116 showed obvious antiviral activity at the post-entry phase; however, no antiviral activity was detected when it was added during the entry phase. The antiviral activity of Remdesivir, as the control group, was also detected in the time-of-addition assay, and the result is comparable to that obtained for VV116. Remdesivir is a known anti-coronavirus nucleoside analog that takes effect at the stage of viral genome replication by viral RdRp. On the other hand, heparan sulfate, another control group, demonstrated antiviral activity in the group where compound addition occurs concurrently with viral infection. The antiviral activity of heparan sulfate was abolished when this compound was added during the post-entry phase. This result is consistent with that of a previous study, which showed heparan sulfate inhibits the attachment and entry of HCoV-NL63 [31]. Based on the above results, we concluded that VV116 exhibits its anti-coronavirus activity through the inhibition of viral RdRp replication. Furthermore, an rNTP competitive inhibition assay was performed. As shown in Figure 3B, the anti-coronavirus activity of VV116 was significantly attenuated in a dose-dependent manner when rNTP was added. When the concentration of VV116 exceeded 100 μM, the antiviral activity of VV116 was almost abolished, and the nucleoprotein expression of HCoV-NL63 returned to a similar level to that of the control group. On the contrary, the addition of rNTP had no effect on the antiviral activity of heparan sulfate since it exerts its function during the stage of early viral invasion. These results further suggest that the mechanism of VV116 anti-coronavirus activity is to inhibit the replication ability of viral RdRp.

### 3.4. VV116 Exhibits Substantial Anti-Coronavirus Activity in an MHV-Infected Mouse Model

The mouse hepatitis virus (MHV) belongs to the coronavirus family and is a naturally occurring virus that infects mice. The MHV-infected mouse model is often used to evaluate the effect of anti-coronavirus drugs. Thus, we used this model to confirm the anti-coronavirus activity of VV116 in vivo. Three-week-old mice were intranasally infected with the virus and then treated with VV116 at different concentrations (25, 50, and 100 mg/kg, bis in die). Mice were sacrificed three days after infection, and liver tissues were collected and subjected to the measurement of viral genome copies and viral titers. As indicated in Figure 4A, both viral genome copies and titers decreased substantially with increasing VV116 concentration. In the experimental group in which the highest compound dose (100 mg/kg) was applied, both viral genome copies and titers decreased by four orders of magnitude. To further confirm the antiviral activity of VV116, histopathology was performed on the liver tissues of mice from each experimental group (Figure 4B). The livers of mice in the control group (mice infected with MHV but not treated with VV116) displayed a high number of focal infiltrations of perivascular inflammatory cells (yellow arrows) as well as a small number of hepatocytes with mild fatty degeneration. Some hepatocytes’ cytoplasms contained minute, spherical vacuoles. Following the administration of VV116, the infiltration of inflammatory cells was reduced. There were scattered inflammatory cell infiltrations around a small number of blood vessels in the livers of mice treated with 25 mg/kg VV116 but no obvious inflammatory cell infiltrations in the livers of mice treated with higher doses of VV116 (50 or 100 mg/kg).

As infection with MHV will lead to death in mice, we also investigated the protective effect of VV116 for mice. Mice were infected with MHV, treated with VV116 at different concentrations (25 mg/kg, 50 mg/kg, and 100 mg/kg), and monitored daily for symptoms of disease, including changes in body weight and death. The results show that more than 80% of mice in the control group (mice infected with MHV but not treated with VV116) died on day 7 post-infection. Mice in the experimental groups (mice infected with MHV and treated with different concentrations of VV116) demonstrated a significant improvement in survival. Furthermore, in the experimental groups treated with VV116 at concentrations of 50 mg/kg and 100 mg/kg, no deaths were found on day 7 post-infection (Figure 4C). Weight loss is a common symptom of MHV infection in mice. Figure 4D shows that VV116 treatment can reduce weight loss, as demonstrated by the comparison with the control group. In the experimental groups treated with VV116 at concentrations of 50 mg/kg and 100 mg/kg, the trend of weight loss ended on day 6 post-infection, and the mice’s weight increased from day 7 onwards. This finding suggests that high-dose VV116 treatment effectively suppressed viral titers in vivo, alleviating weight loss caused by viral infections. Overall, our results confirm that VV116 can exhibit substantial anti-coronavirus activity in vivo, protecting mice from weight loss and death caused by viral infection.

## 4. Discussion

In this study, we systematically evaluated the antiviral effect of VV116 against various coronaviruses, including low-pathogenic human coronaviruses such as HCoV-NL63, HCoV-229E, and HCoV-OC43 and animal coronaviruses such as MHV, FIPV, FECV, and CCoV. We found that VV116 has substantial antiviral activity against these coronaviruses, especially against feline and canine coronaviruses (with an EC_50_ between 0.665 and 0.847 μM). The human coronavirus HCoV-NL63 also showed an obvious inhibitory response to VV116 (with an EC_50_ of 1.087 μM). Interestingly, it was found that the antiviral activity of VV116 against HCoV-NL63 is more significant at the protein level than that at the RNA level. An explanation for this could be that viral protein expression occurs downstream of viral RNA expression and the difference in viral quantity among different experimental groups is further amplified as it travels downstream.

Since the outbreak of COVID-19 in 2020, much research has been focused on the infection and epidemic transmission of SARS-CoV-2; however, other coronaviruses are still at large. Low-pathogenic human coronaviruses are prevalent worldwide and can severely impact people’s daily lives. Infection with these viruses can lead to upper respiratory diseases, and it is reported that they cause 15–30% of common cold symptoms in adults [32]. Moreover, infections in immunodeficiency patients and children under 5 years old may lead to fatal pneumonia or bronchitis. Humans cannot obtain long-lasting protective immunity against these viruses, resulting in a high probability of reinfection, which further illustrates the importance of developing broad-spectrum antiviral drugs against these coronaviruses. Judging from current trends, we expect the future will be one in which SARS-CoV-2 coexists alongside other low-pathogenic human coronaviruses. Developing a broad-spectrum anti-coronavirus drug with a significant inhibitory effect on both SARS-CoV-2 and other low-pathogenic human coronaviruses would thus present a significant advantage for human health and epidemic prevention.

Animal coronaviruses are more harmful to the host than low-pathogenic human coronaviruses. For example, FIPV infection can lead to feline infectious peritonitis, which is a highly fatal viral disease in cats. Furthermore, CCoV infection can cause gastroenteritis symptoms of varying severity and seriously affect the health of dogs. The health of cats and dogs, as the most common companions of humans, has attracted much attention. A broad-spectrum drug that can effectively inhibit both human and animal coronaviruses would greatly improve the quality of life of animals and the well-being of humans. In addition, it should be noted that coronaviruses are RNA genome viruses with a high frequency of mutation and recombination, which makes them inherently prone to and capable of cross-species transmission across natural biological barriers, allowing them to adapt and survive quickly in new environments. There have been many cases of this throughout history. For example, HCoV-OC43 is the result of the bovine-to-human transmission of bovine coronavirus around 1890 [33], and HCoV-229E is suggested to be the descendant of camelid-associated viruses [34]. SARS-CoV emerged in 2003, MERS-CoV emerged in 2012, and SARS-CoV-2 emerged in 2020. These are all coronaviruses that originated in bats and can infect humans through intermediate spillover hosts such as civet cats, dromedary camels, and pangolins [35]. Cats and dogs are animals in close contact with humans in daily life, and the coronaviruses they carry also have a high probability of being transmitted to humans due to mutations. There have been cases of feline coronavirus and canine coronavirus found in human samples [36,37], reminding us of the threat of the cross-species transmission of feline coronavirus and canine coronavirus. The development of a drug that can inhibit both human and animal coronaviruses is thus not only important for the protection of the animals around us but also ourselves.

In summary, we demonstrated that VV116, as a nucleoside analog proven to inhibit the RdRp of SARS-CoV-2 by others, exhibited broad-spectrum anti-coronavirus activity in this study, which is consistent with the fact that the coronavirus RdRp is relatively highly conserved. Our work provides a comprehensive and systematic understanding of the ability of VV116 to fight against coronaviruses and supports the expansion of subsequent clinical applications of VV116 with detailed experimental data. Drugs with broad-spectrum anti-coronavirus activity, on the one hand, are beneficial for the simultaneous treatment and prevention of multiple coronavirus infections, and on the other hand, provide promising candidate drugs for combatting future coronavirus outbreaks, buying precious time to fight against epidemics.

## Figures and Tables

**Figure 1 viruses-15-02295-f001:**
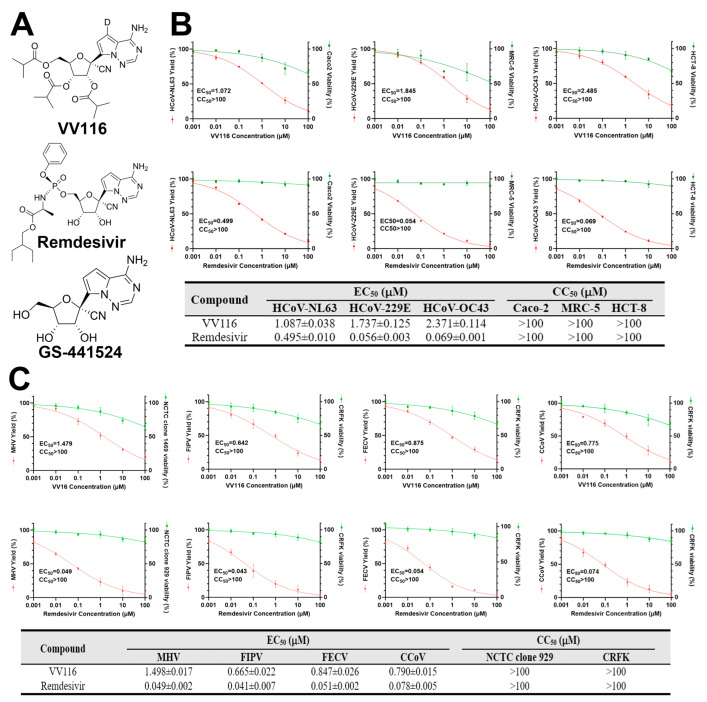
Antiviral activity of VV116 and Remdesivir against human and animal coronaviruses. (**A**) The chemical structures of VV116, Remdesivir, and GS-441524. GS-441524 is a well-studied broad-spectrum antiviral agent that inhibits RNA-dependent RNA polymerase (RdRp). Remdesivir is the phosphoramidate of the parent nucleoside GS-441524 and can be metabolized to GS-441524 triphosphate in the cell. VV116 is a deuremidevir isadeuterated, tris-isobutyric acid ester prodrug of GS-441524 with good oral bioavailability and is often referred to as a Remdesivir derivative. (**B**) The upper panel depicts a dose–response study on VV116 and Remdesivir against human coronaviruses (HCoV-NL63, HCoV-229E, and HCoV-OC43) in host cells (Caco-2, MRC-5, and HCT-8). In the lower panel, the half-maximal effective concentration (EC_50_) and the mean 50% cytotoxic concentration (CC_50_) of VV116 and Remdesivir are displayed. (**C**) The upper panel depicts a dose–response study on VV116 and Remdesivir against animal coronaviruses (MHV, FIPV, FECV, and CCoV) in host cells (NCTC clone 929 and CRFK). In the lower panel, the half-maximal effective concentration (EC_50_) and the mean 50% cytotoxic concentration (CC_50_) of VV116 and Remdesivir are displayed.

**Figure 2 viruses-15-02295-f002:**
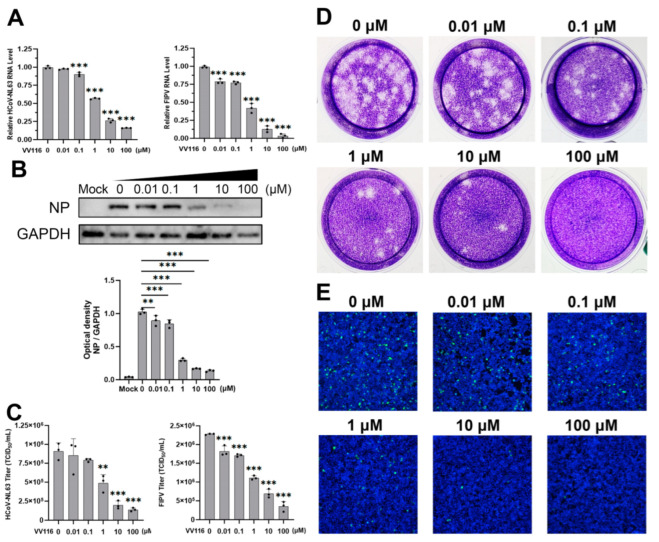
Confirmation of the effect of VV116 on the replication of HCoV-NL63 and FIPV. Cells were infected with HCoV-NL63 or FIPV and then treated with different concentrations of VV116. Dose-dependent inhibition of HCoV-NL63 or FIPV replication by VV116 treatment was confirmed through the replication of HCoV-NL63 and FIPV detected in the form of relative RNA levels (**A**) and the virus titers in the supernatant (**C**). (**B**) In addition, the dose-dependent inhibition of HCoV-NL63 replication by VV116 was revealed through Western blot. Representative Western blots and bar graphs show the expression and densitometric ratios of NP to GAPDH. (**D**) The virus titers in the supernatant were detected using a plaque assay. (**E**) The dose-dependent inhibition of HCoV-NL63 replication by VV116 was also revealed using an immunofluorescence assay. Caco-2 cells were infected with HCoV-NL63, and the various indicated concentrations of VV116 were added 1 h post-infection. Twenty-four hours after infection, cells were fixed and permeabilized. The virus was detected with HCoV-NL63 nucleoprotein antibody and FITC-conjugated secondary antibody, and the nucleus was stained with DAPI. ** *p* < 0.01 versus control, *** *p* < 0.001 versus control.

**Figure 3 viruses-15-02295-f003:**
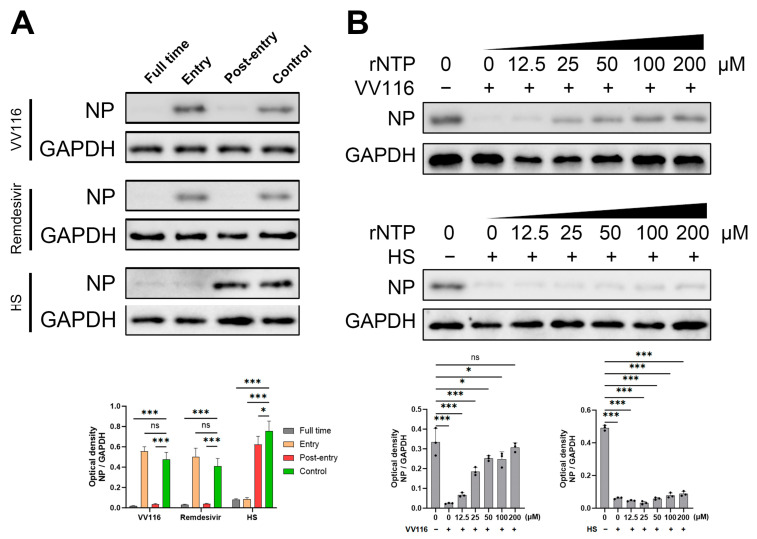
VV116 exerts antiviral activity by inhibiting the RdRp activity of HCoV-NL63. (**A**) A time-of-addition assay revealed that VV116 functions during the viral genome replication stage. (**B**) An rNTP competitive inhibition assay was conducted to confirm that VV116 inhibited RdRp activity. Representative Western blots and bar graphs show the expression and densitometric ratios of NP to GAPDH in different experimental groups. ns *p* > 0.05 versus control, * *p* < 0.05 versus control, *** *p* < 0.001 versus control.

**Figure 4 viruses-15-02295-f004:**
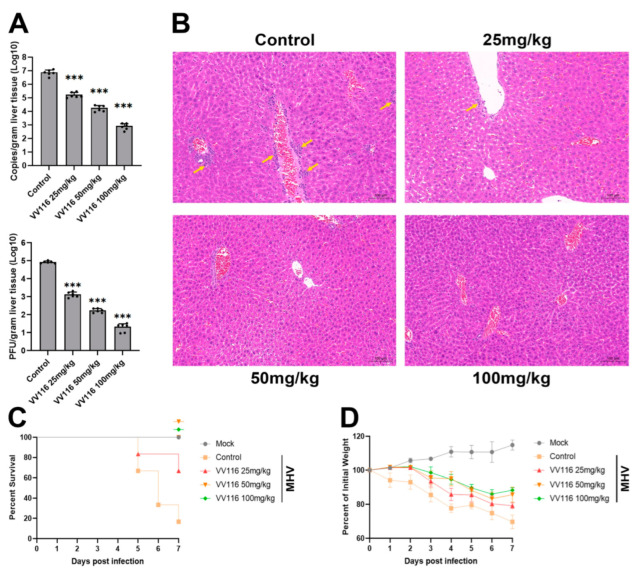
VV116 exhibits its antiviral activity in an MHV-infected mouse model. (**A**) MHV-infected mice were treated with different concentrations of VV116. Three days after infection, the viral genome copies and viral titers in the livers of the mice in each experimental group were measured. *** *p* < 0.001 versus control. (**B**) Representative liver tissue histopathology of mice in each experimental group. Liver slices were stained with hematoxylin and eosin. In the absence of VV116 treatment, a large number of inflammatory cell infiltrations (yellow arrows) were observed in the liver of MHV-infected mice. As the VV116 concentration increased, the infiltration of inflammatory cells decreased. The survival ratio (**C**) and body weight (**D**) of MHV-infected mice treated with vehicle or varying concentrations of VV116 were monitored daily for seven days after infection.

**Table 1 viruses-15-02295-t001:** Primers used in this study.

Primers	Sequence (5′ to 3′)
NL63 qPCR-F	GTGATGCATATGCTAATTTG
NL63 qPCR-R	CTCTTGCAGGTATAATCCTA
FIPV qPCR-F	CCGAGGAATTACTGGTCATCGCG
FIPV qPCR-R	GCTCTTCCATTGTTGGCTCGTC
GAPDH qPCR-F	AAGGCTGTGGGCAAGG
GAPDH qPCR-R	TGGAGGAGTGGGTGTCG

## Data Availability

All data and materials used in this study are included in the published article.

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
