# Peer review of "Remdesivir Derivative VV116 Is a Potential Broad-Spectrum Inhibitor of Both Human and Animal Coronaviruses"

_viruses, 2023, doi:10.3390/v15122295_

Round 1

Reviewer 1 Report

Comments and Suggestions for Authors

The Article is not a great novelty in the Coronavirus research, there is a tentative to find some indications for a new molecule.

Here are some suggestions.

1. In the title substitute the word "potent" with "as a new" or a "potential".

2. Please include a Figure with chemical structures of all the drugs and pro-drugs and derivatives mentioned in the study, and also a general scheme of action of Remdesivir and derivatives.

3. There are many important articles and reviews about Remdesivir available with a simple search on PubMed; why no one are mentioned in the references of the articles. Choose someone to increase appropriate cited refs.

4. Please check the description of the experiments in Materials and Methods. There are some flaws, like replicates, and statistics, rNTP are not mentioned.

5. Authors should put the Figure Captions after the Figures and not before.

6. Erase the sentence from line 172 to line 174.

7. The results with semilog graphs are not correct for extrapolation of EC50, in fact you are an absurd comparison of decimals (.xxx uM) when data are reported. Authors carefully re-elaborate this point and please insert the number of replicates, in many points there aren't standard deviations.

8. In Figure 2 there is no description in the caption of panels D and E (please add E in the figure), please insert arrows to help the description.

9. The inhibition assay is a competition of a sort of binding displacement not an enzymatic experiments of competitive inhibition. Authors need to choose or re-make the experiment or explain and describe better what their are thinking. 

10. Figure 4A in Y-axis with Log is misleading about the real quantity of mg/kg nearby the control that are necessary to reduce the copies/gram liver tissue. Figure 4D is not mentioned in the text, probably there are confusion in the results when are reported the last panels. Finally, why you stop at day 7.

Comments on the Quality of English Language

The english is good but there are redundant repetitions like "obvious" and "sistematically".

Author Response

Thank you very much for spending your valuable time and effort reviewing our manuscript. We have revised the manuscript based on your constructive comments and major changes are highlighted in yellow. In addition, we used the MDPI editing services to polish the manuscript. The amendments are detailed below.

Comments 1. In the title substitute the word "potent" with "as a new" or a "potential".

Response 1. Thanks for your suggestion, “potent” has been changed to “potential” in the revised manuscript.

Comments 2. Please include a Figure with chemical structures of all the drugs and pro-drugs and derivatives mentioned in the study, and also a general scheme of action of Remdesivir and derivatives.

Response 2. Thank you for your suggestion. We have added the structure diagrams and related descriptions of Remdesivir and VV116 to the manuscript (Figure 1A and line 92-97).

Comments 3. There are many important articles and reviews about Remdesivir available with a simple search on PubMed; why no one are mentioned in the references of the articles. Choose someone to increase appropriate cited refs.

Response 3. The original intention of this research is to study the inhibitory effect and possible mechanisms of VV116 on various coronaviruses other than SARS-CoV-2. Remdesivir is a well-known antiviral nucleoside analog, and it was used as a positive control in this research, so the antiviral function of Remdesivir is not specifically introduced in the introduction part. Thank you for your reminder. We have realized that the lack of this part may cause some background confusion for readers, so we have added the relevant introduction content and literature citations (line 63-70).

Comments 4. Please check the description of the experiments in Materials and Methods. There are some flaws, like replicates, and statistics, rNTP are not mentioned.

Response 4. Thank you for your reminder. We have supplemented and improved the Materials and Methods section in the revised manuscript.

Comments 5. Authors should put the Figure Captions after the Figures and not before.

Response 5. Thank you for your reminder, it has been corrected in the manuscript.

Comments 6. Erase the sentence from line 172 to line 174.

Response 6. Thank you for your reminder, it has been corrected in the revised manuscript.

Comments 7-1. The results with semilog graphs are not correct for extrapolation of EC50, in fact you are an absurd comparison of decimals (.xxx uM) when data are reported.

Response 7-1. Thank you for pointing out the problem. We have changed the compound concentration values to normal decimal values in Figure 1.

Comments 7-2. Authors carefully re-elaborate this point and please insert the number of replicates, in many points there aren't standard deviations.

Response 7-2. Each point in each graph in Figure 1 corresponds to the average value of three replicates. For some points, the error bars cannot be seen. That is because the error bar is shorter than the size of the symbol, so the GraphPad Prism won't draw it. The same question is also answered on the official website of GraphPad Prism (https://www.graphpad.com/support/faq/why-are-there-no-error-bars-on-some-points/). In addition, we have made the symbol of each point in the figure smaller to make the error bars more visible.

Comments 8. In Figure 2 there is no description in the caption of panels D and E (please add E in the figure), please insert arrows to help the description.

Response 8. Thank you for pointing out the problem. We have rewritten the figure legend and added E in the Figure (line 194-202). Because our results are presented as the changes in the number of plaques (Figure 2D) or fluorescent spots (Figure 2E) in the entire field of view, and the use of arrow marks only emphasizes a certain part of the field of view, we did not add arrow marks in the end.

In order to improve readability, we have also made the following modifications: (1) The direction of error bars is uniformly set to upward in Figure 2A and 2C; (2) The VV116 concentration annotations were added to each plaque image in Figure 2D; (3) The image brightness and contrast were enhanced in Figure 2E.

Comments 9. The inhibition assay is a competition of a sort of binding displacement not an enzymatic experiments of competitive inhibition. Authors need to choose or re-make the experiment or explain and describe better what their are thinking.

Response 9. You are right. The results in Figure 3 can only directly prove that the inhibitory effect of VV116 on HCoV-N63 is in the viral replication stage and can be interfered by rNTP, implying that VV116 is probably incorporated into the newly synthesized viral RNA genome in the form of nucleic acid analogs. Because VV116 has been confirmed to inhibit SARS-CoV-2 through this mechanism, combined with our current results, we speculate that its inhibitory mechanism for HCoV-NL63 is likely to be the same.

Originally, we wanted to directly prove this mechanism through in vitro biochemical experiments, which required the use of 116-NTP (the nucleoside triphosphate form of VV116). For VV116 to have an antiviral effect in cells, it must first be phosphorylated by intracellular kinases into its nucleoside triphosphate form, 116-NTP. In in vitro biochemical experiments, since there are no intracellular kinases, 116-NTP needs to be used directly. Because there is no commercial product of this compound at present, we cannot obtain it through purchase, so we cannot complete this experiment, which is indeed a pity.

Comments 10-1. Figure 4A in Y-axis with Log is misleading about the real quantity of mg/kg nearby the control that are necessary to reduce the copies/gram liver tissue.

Response 10-1. Sorry, I don’t quite understand what you mean about the first question. Do you mean that the display form of the Y-axis value is inappropriate? This kind of presentation has been used in many articles, such as Figure 1E in the article by Zhang et al. [1], or Figure 6C in the article by Zhao et al [2].

Comments 10-2. Figure 4D is not mentioned in the text, probably there are confusion in the results when are reported the last panels.

Response 10-2. Regarding the second question, it is actually due to a clerical error. In the original text, Figure 4B in line 309 should be Figure 4C, and Figure 4C in line 310 should be Figure 4D. It has been corrected in the revised version (line 357-358), and I'm sorry for causing trouble to your reading.

Comments 10-3. Finally, why you stop at day 7.

Response 10-3. Regarding the third question, it is true that the general experiment of mouse weighing lasts 14 days after viral infection. There are two reasons why we only did it for 7 days. First, based on our results, we found that there were already obvious differences between experimental groups within 7 days after viral infection, so the experiment was not continued. Second, there are also many studies that end at around 7 days after viral infection, such as Figure 5D in the article by Zhao et al. [2], Figure 3A and 3B in the article by Tran et al. [3], and Figure 1A in the article by Khanolkar et al [4].

Once again, we sincerely thank you for your valuable comments. We benefited a lot.

Reference

  1. Zhang, R.; Zhang, Y.; Zheng, W.; Shang, W.; Wu, Y.; Li, N.; Xiong, J.; Jiang, H.; Shen, J.; Xiao, G.; et al. Oral Remdesivir Derivative VV116 Is a Potent Inhibitor of Respiratory Syncytial Virus with Efficacy in Mouse Model. Signal Transduct Target Ther 2022, 7, 123, doi:10.1038/s41392-022-00963-7.
  2. Zhao, Z.; Xiao, Y.; Xu, L.; Liu, Y.; Jiang, G.; Wang, W.; Li, B.; Zhu, T.; Tan, Q.; Tang, L.; et al. Glycyrrhizic Acid Nanoparticles as Antiviral and Anti-Inflammatory Agents for COVID-19 Treatment. ACS Appl. Mater. Interfaces 2021, 13, 20995–21006, doi:10.1021/acsami.1c02755.
  3. Tran, V.; Moser, L.A.; Poole, D.S.; Mehle, A. Highly Sensitive Real-Time In Vivo Imaging of an Influenza Reporter Virus Reveals Dynamics of Replication and Spread. J Virol 2013, 87, 13321–13329, doi:10.1128/JVI.02381-13.
  4. Khanolkar, A.; Hartwig, S.M.; Haag, B.A.; Meyerholz, D.K.; Harty, J.T.; Varga, S.M. Toll-Like Receptor 4 Deficiency Increases Disease and Mortality after Mouse Hepatitis Virus Type 1 Infection of Susceptible C3H Mice. Journal of Virology 2009, 83, 8946–8956, doi:10.1128/JVI.01857-08.

Reviewer 2 Report

Comments and Suggestions for Authors

The study by Liu et al., entitled: Remdesivir derivative VV116 is a potent broad-spectrum inhibitor against both human and animal coronaviruses, is interesting. However, the study have a huge scope of improvement. 

First, what is the significance of the study? As per figure 1, the antiviral activity of VV116 is at a significantly low (inhibits virus at significantly high concentration), whereas there are multiple studies on the chemicals/drugs that has broad spectrum antiviral activity at a significantly lower concentrations. 

Tables should have separate heading and appropriate citation in the text.

How many times the western blots were performed. Authors should include densitometric analysis (bar graphs) of western blot images.

Authors should include combination experiments of VV116 and Remdesivir. 

Including the structure of VV116 in the text will be very helpful. 

Line 60/61 needs reference/s. 

Author Response

Thank you very much for spending your valuable time and effort reviewing our manuscript. We have revised the manuscript based on your constructive comments and major changes are highlighted in yellow. In addition, we used the MDPI editing services to polish the manuscript. The amendments are detailed below.

Comments 1. First, what is the significance of the study? As per figure 1, the antiviral activity of VV116 is at a significantly low (inhibits virus at significantly high concentration), whereas there are multiple studies on the chemicals/drugs that has broad spectrum antiviral activity at a significantly lower concentrations.

Response 1. The significance of this study is to prove that VV116 has broad-spectrum anti-coronavirus activity and support for its subsequent expansion of therapeutic applications with experimental data. Although its anti-coronavirus effect is not particularly excellent, the advantage is that it is safe and can be taken orally. We have demonstrated that oral VV116 can effectively resist MHV infection through mouse experiments, which suggests that it is quite possible to treat animals infected with coronavirus (especially cats, dogs and other pets that are in close contact with humans) through oral administration in the future.

Comments 2. Tables should have separate heading and appropriate citation in the text.

Response 2. Thank you for your reminder. Because according to the requirements of the template document, "Tables should be placed in the main text near to the first time they are cited.", and the first mention of primers in our article is in "2.4 RNA extraction and real-time PCR", so we placed Table 1 at the end of this paragraph.

Comments 3. How many times the western blots were performed. Authors should include densitometric analysis (bar graphs) of western blot images.

Response 3. Each western blot was repeated at least three times independently, and we have included the densitometric analysis histograms in the results based on your suggestion. Thank you for your reminder.

Comments 4. Authors should include combination experiments of VV116 and Remdesivir.

Response 4. Thank you for your suggestion. Because the purpose of our research is to study the anti-coronavirus activity of VV116, and Remdesivir is only used as a control group, so we have not conducted experiments like this.

Comments 5. Including the structure of VV116 in the text will be very helpful.

Response 5. Thank you for your suggestion. We have added the structure diagrams and related descriptions of Remdesivir and VV116 to the manuscript (Figure 1A and line 92-97).

Comments 6. Line 60/61 needs reference/s.

Response 6. Thank you for your reminder. We have cited some references here to further facilitate readers to understand the relevant background information (line 72-74).

Once again, we sincerely thank you for your valuable comments. We benefited a lot.

Reviewer 3 Report

Comments and Suggestions for Authors

In this manuscript Liu et al. analyzed the antiviral activities of the Remdesivir derivative VV116 on a broad spectrum of endemic human and animal coronaviruses. Phase III clinical trials using VV116 have been conducted in SARS-CoV-2 infected patients, a fact the authors decided to completely ignore throughout their manuscript. No single publication about the preclinical and clinical development of VV116 was cited. If they had done so, they would had admit the redundancy of their experimental work. Even worse, the comparison of their own results with those of these studies would unravel the discrepancies and experimental short falls of their work.   

On the positive side, the authors demonstrate for the first time a broad antiviral efficacy of VV116 against various human and animal coronaviruses, especially with regard to pandemic preparedness and the fact that VV116, in contrast to Remdesivir, is orally available. The new findings of this manuscript can be summarized as: i) VV116 is effective against other human coronaviruses besides SARS-CoV-2, and ii) in vivo VV116 prevent infection and disease by MHV. The entire manuscript should be reduced onto these 2 points. The analysis of the authors regarding the mode of action are redundant in the light of already published data and should be removed from the manuscript.

Furthermore, the manuscript contains countless grammatical, spelling, expression, and content errors that need to be removed. It´s sloppy and premature submission style represents a deliberated depreciation to the reviewers.

Thus, the manuscript is not suitable for publication as it stands now and needs major revisions.

Major points

1. The abstract requires complete rewriting. As it stands now the abstract contains only very poor information about the experiments performed and also the different virus systems used. Furthermore, the results are not clearly presented. The authors write in line 15: "...and confirmed its broad-spectrum anti-coronavirus activity.". This would mean that they only “confirm” already known and published results. Is this true or do they show here for the first time a braod-spectrum antiviral activity against different coronaviruses. Furthermore, the abstract contains many grammatical and phrasing errors, as well as errors in content.

2. The authors must summarize the clinical data on the development of VV116 in China. In the current manuscript, they leave the readers in the dark about the successful clinical development of VV116. Everything about the tox, PK and PD was done in the preclinical development. Thus, there was nothing new to discover about those issues. Therefore, the authors must compare their virus inhibition results with the comprehensive data sets from preclinical and clinical trial, especially regarding the plasma concentrations and EC50 values.

3. The analysis of the authors regarding the mode of action by performing time of addition experiments are senseless and should be removed for different reasons: i) it is written in stone and published in countless papers that Remdesivir blocks the activity of the viral RNA-dependent RNA-polymerase (RdRP). VV116 is also already known to inhibit RpRP. Therefore, these results are far from unexpected and do not provide any novelty benefit. Especially because the authors decided to perform these experiments only with one of the coronaviruses they used in Figure 1 (HCoV-NL63). They conclude from these results that the same is true for all other Coronaviruses used in this study. However, this could also have been concluded from the already published data on the inhibition of RdRP after SARS-Cov-2 infection.

ii) By using the experimental setup of the time of addition experiments, the authors cannot conclude that the RdRP is inhibited. They can only conclude that the inhibitor intervenes in post entry events. In this context, it would have been interesting to check whether the RdRP of the different human and animal coronaviruses used in this study are the similar or different in sequence and/or structure and whether VV116 inhibits RdRP activity with different efficacy. For this purpose, in vitro activity assays should have been performed with the different RdRP. Such results would have presented real novelty value and would have been very interesting.    

4. Throughout the manuscript, there are countless passages and sentences that make no sense at all, or whose message is incorrect. Furthermore, there are countless grammar and spelling mistakes that disturb the flow of reading. It is not possible to read the manuscript at all, due to the break of flow within some paragraphs. To list all of these mistakes would go far beyond the scope of this review, so I will only list a few of them here.  

-line 11 and line 23: “Both hu-10 man and animal coronaviruses are closely related to humans…” and “They are closely related to humans be-23 cause their infections frequently cause respiratory diseases.” Viruses and humans are NOT related!

- line 34: “The fatal rates of SARS-CoV and MERS-CoV are 10% and 3.4%,…”. The fatal rate of MERS is around 30% and not 3.4%.

- line 45: “…and there are two virulent biotypes of feline coronavirus in…”. “Biotype” is not a virological term.

- line 155: “…with VV116 orally bis in die (25 mg/kg, 50 mg/kg, and 100 mg/kg).” What do the authors mean with “bis in die”?

- line 172-174: “This section may be divided by subheadings. It should provide a concise and precise 172 description of the experimental results, their interpretation, as well as the experimental 173 conclusions that can be drawn.” This text originates from the journal template and should preferably be removed before the review process. This sloppiness demonstrate the disrespect of the authors to the reviewers.

- line 208-213: “When the concentration of VV116 was 0.1 μM, the viral RNA expression of  HCoV-NL63 had already been reduced with a statistically significant difference compared  with the control group. When the concentration of VV116 reached 1 μM, the viral RNA  expression was half that of the control group. When the concentration of VV116 exceeded 10 μM, the viral RNA expression was less than a quarter of that of the control group (Figure 2A left panel).

This is just an example of a text passage that appears repeatedly in the results section in a similar way. Such detailed descriptions of the effect at each concentration of VV116 used should be clearly condensed and summarized in one sentence each.

- Figure 2 D: the figure is not labeled. Figure 2 E: the figure contains no “E”. Even worse, the quality and significance of the figure 2 E is inacceptable. The reader can hardly recognize infected cells and differences between the used concentrations of VV116 are not detectable. Therefore, this subfigure should be completely removed.

- Figure 3: In the figure legend, the authors address a Figure 3 C. Where is this figure?

- line 328: “We found that VV116 has substantial antiviral activity against the above viruses,…”. “Above viruses” is no scientific term and must be specified.

- line 371-379: this section has a different font size

The examples listed above were just a handful of many other such errors by the authors. Thus, the article should be completely revised by a native speaker.

5. Figure 3 B: In this figure the authors only show one Western blot in which no differences can be seen, especially in the low concentration ranges of VV116. In order to obtain statistical significance regarding these experiments, the authors must carry out a densitometric evaluation of all Western blots performed; otherwise, this figure has no significance whatsoever.

6. The results of Figure 1 show that the EC50 values of VV116 are in some cases 1-2 log stages higher than these from Remdesivir. However, the authors do not mention or even discuss this discrepancy. Do they have an explanation for this observation? They should also compare their results with the published results for the EC50 of VV116 and Remdesivir regarding SARS-CoV-2.

Minor points

1. The authors compare VV116 and Remdesivir in their study. However, they never mention the chemical differences of these two compounds. It would also be very helpful for the reader when they depict a chemical structure of these two compounds.

2. Figure 1 A need revision. The pictures and especially the lettering are too small and the reader cannot recognize anything.

3. Line 225-227: “The explanation could be that viral protein expression occurs downstream of viral RNA expression and that the difference in viral quantity among different experimental groups is further amplified as it goes downstream.” Speculations must be shifted into the discussion section.

4. line 371: “Collectively, we demonstrated that VV116, as a nucleoside analog targeting the coronavirus RdRp,…”. This sentence must be removed, as the authors did not show in this manuscript, that VV116 directly targets and inhibits the RdRP.

Comments on the Quality of English Language

The article should be completely revised by a native speaker. For specific comments see "Comments and Suggestions for Authors
".

Author Response

Thank you very much for spending your valuable time and effort reviewing our manuscript. We have revised the manuscript based on your constructive comments and major changes are highlighted in yellow. In addition, we used the MDPI editing services to polish the manuscript. The amendments are detailed below.

Comments 1-1. The abstract requires complete rewriting. As it stands now the abstract contains only very poor information about the experiments performed and also the different virus systems used. Furthermore, the results are not clearly presented.

Response 1. The abstract of the article has been rewritten to highlight the experimental details and results of this study (line 13-26).

Comments 1-2. The authors write in line 15: "...and confirmed its broad-spectrum anti-coronavirus activity.". This would mean that they only “confirm” already known and published results. Is this true or do they show here for the first time a braod-spectrum antiviral activity against different coronaviruses.

Response 1-2. I am sorry that our wording has caused ambiguity for you. To our knowledge, it is the first time that the antiviral effects of VV116 against low-pathogenic human coronaviruses (HCoV-NL63, HCoV-229E, and HCoV-OC43) and animal coronaviruses (MHV, FIPV, FECV, and CCoV) was studied. The word “confirm” in the original abstract in line 15 was to convey a meaning that we proved it’s to be true that VV116 has a broad-spectrum anti-coronavirus activity. Since the abstract has been rewritten, this ambiguity disappeared.

Comments 1-3. Furthermore, the abstract contains many grammatical and phrasing errors, as well as errors in content.

Response 1-3. The abstract has been rewritten, and the revised manuscript was then polished by MDPI editing services.

Comments 2-1. The authors must summarize the clinical data on the development of VV116 in China. In the current manuscript, they leave the readers in the dark about the successful clinical development of VV116.

Response 2-1. Thank you for the reminder. We realize that the lack of information on clinical studies of VV116 may cause confusion for readers, so we have added relevant information in the revised manuscript (line 72-85).

Comments 2-2. Everything about the tox, PK and PD was done in the preclinical development. Thus, there was nothing new to discover about those issues. Therefore, the authors must compare their virus inhibition results with the comprehensive data sets from preclinical and clinical trial, especially regarding the plasma concentrations and EC50 values.

Response 2-2. The focus of this research is to prove that VV116 has a good antiviral effect against various coronaviruses other than SARS-CoV-2. In addition, taking HCoV-NL63 as a representative, we also tried to investigate the antiviral mechanism of VV116 to see if it is consistent with the anti-SARS-CoV-2 mechanism. Therefore, this study did not involve comparing the differences in the antiviral effects of VV116 against SARS-CoV-2 and other coronaviruses, nor does it involve comparing the antiviral effects of VV116 with other compounds.

Comments 3-1. The analysis of the authors regarding the mode of action by performing time of addition experiments are senseless and should be removed for different reasons: i) it is written in stone and published in countless papers that Remdesivir blocks the activity of the viral RNA-dependent RNA-polymerase (RdRP). VV116 is also already known to inhibit RpRP. Therefore, these results are far from unexpected and do not provide any novelty benefit.

Response 3-1. Although the mechanism by which VV116 inhibits the replication of the SARS-CoV-2 has been thoroughly studied, and it is highly likely that the mechanism by which it inhibits other coronaviruses is similar, we believe that this study still has certain significance. At least we have directly proved this speculation through experimental data and further deepened people's understanding of VV116's anti-coronavirus mechanism.

Comments 3-2. Especially because the authors decided to perform these experiments only with one of the coronaviruses they used in Figure 1 (HCoV-NL63). They conclude from these results that the same is true for all other Coronaviruses used in this study. However, this could also have been concluded from the already published data on the inhibition of RdRP after SARS-Cov-2 infection.

Response 3-2. When conducting research on the anti-coronavirus mechanism of VV116, we originally planned to use HCoV-NL63 and FIPV as representatives of human and animal coronaviruses. Since there are no commercial FIPV antibodies available, only HCoV-NL63 was studied in western blot experiments (including time of addition).

As you said, a single HCoV-NL63 investigation cannot lead to the conclusion that the mechanism of VV116 inhibiting all coronaviruses is the same. We just hope to add some new data to the existing research on the antiviral mechanism of VV116. Based on our experimental results and the research of others, we can speculate that it is highly likely that the mechanism by which VV116 inhibits other coronaviruses is the same. However, for each specific coronavirus, it can only be said to be true after it is proved through experiments.

Comments 3-3. ii) By using the experimental setup of the time of addition experiments, the authors cannot conclude that the RdRP is inhibited. They can only conclude that the inhibitor intervenes in post entry events. In this context, it would have been interesting to check whether the RdRP of the different human and animal coronaviruses used in this study are the similar or different in sequence and/or structure and whether VV116 inhibits RdRP activity with different efficacy. For this purpose, in vitro activity assays should have been performed with the different RdRP. Such results would have presented real novelty value and would have been very interesting.

Response 3-3. Thanks for your advice. We did pay attention to this problem when designing the experiment, so the best way should be to directly prove the inhibitory effect of VV116 on the RdRp activity of HCoV-NL63 through in vitro biochemical experiments without interference from cellular contents. For VV116 to have an antiviral effect in cells, it must first be phosphorylated by intracellular kinases into its nucleoside triphosphate form, 116-NTP. In in vitro biochemical experiments, since there are no intracellular kinases, 116-NTP needs to be used directly. Because there is no commercial product of this compound at present, we cannot obtain it through purchase, so we cannot complete this experiment, which is indeed a pity.

Comments 4-1. Throughout the manuscript, there are countless passages and sentences that make no sense at all, or whose message is incorrect. Furthermore, there are countless grammar and spelling mistakes that disturb the flow of reading. It is not possible to read the manuscript at all, due to the break of flow within some paragraphs. To list all of these mistakes would go far beyond the scope of this review, so I will only list a few of them here. 

-line 11 and line 23: “Both hu-10 man and animal coronaviruses are closely related to humans…” and “They are closely related to humans be-23 cause their infections frequently cause respiratory diseases.” Viruses and humans are NOT related!

Response 4-1. I am sorry that our wording has caused ambiguity for you. The original meaning of this sentence is that these viruses are closely related to human life, and we or our pets (cats and dogs) can be frequently infected by them. Since the abstract has been rewritten, this ambiguity disappeared.

Comments 4-2. - line 34: “The fatal rates of SARS-CoV and MERS-CoV are 10% and 3.4%,…”. The fatal rate of MERS is around 30% and not 3.4%.

Response 4-2. According to the reference we cited [1] in the original manuscript, the global fatal rate of MERS is 34.56%. It is now corrected in the revised manuscript (line 42). We now suspect that it is confused with the initial 3.4% fatal rate of the SARS-CoV-2 in line 35 in the original manuscript. I am sorry for making such a stupid mistake.

Comments 4-3. - line 45: “…and there are two virulent biotypes of feline coronavirus in…”. “Biotype” is not a virological term.

Response 4-3. Thank you for your reminder. Because the word “Biotype” is often used to classify FECV and FIPV [2–4], we used this word in the original manuscript. In order to make our writing more rigorous and professional, it was replaced by “virulence variant” in the revised manuscript (line 52).

Comments 4-4. - line 155: “…with VV116 orally bis in die (25 mg/kg, 50 mg/kg, and 100 mg/kg).” What do the authors mean with “bis in die”?

Response 4-4. According to Wikipedia, “bis in die” or “b.i.d.” is an abbreviation frequently used in medical prescriptions, meaning twice a day in Latin. This abbreviation is also used in academic papers, such as the article by Coilly et al [5] and article by Zhang et al [6].

Comments 4-5. - line 172-174: “This section may be divided by subheadings. It should provide a concise and precise 172 description of the experimental results, their interpretation, as well as the experimental 173 conclusions that can be drawn.” This text originates from the journal template and should preferably be removed before the review process. This sloppiness demonstrate the disrespect of the authors to the reviewers.

Response 4-5. We sincerely apologize for any offense you may have experienced due to our sloppiness in the preparation of the manuscript. It is now removed from the revised manuscript.

Comments 4-6. - line 208-213: “When the concentration of VV116 was 0.1 μM, the viral RNA expression of  HCoV-NL63 had already been reduced with a statistically significant difference compared  with the control group. When the concentration of VV116 reached 1 μM, the viral RNA  expression was half that of the control group. When the concentration of VV116 exceeded 10 μM, the viral RNA expression was less than a quarter of that of the control group (Figure 2A left panel).”

This is just an example of a text passage that appears repeatedly in the results section in a similar way. Such detailed descriptions of the effect at each concentration of VV116 used should be clearly condensed and summarized in one sentence each.

Response 4-6. Thank you for your suggestions. We have made improvements in the revised version of the manuscript, and it was then polished by MDPI editing services.

Comments 4-7. - Figure 2 D: the figure is not labeled.

Response 4-7. I am sorry for missing the label for Figure 2D. It has been added in the revised manuscript.

Comments 4-8. Figure 2 E: the figure contains no “E”. Even worse, the quality and significance of the figure 2 E is inacceptable. The reader can hardly recognize infected cells and differences between the used concentrations of VV116 are not detectable. Therefore, this subfigure should be completely removed.

Response 4-8. We have rewritten the figure legend and added E in the Figure (line 194-202). In order to improve readability, the image brightness and contrast were enhanced in Figure 2E. The inhibitory effect of VV116 on HCoV-NL63 can be visually displayed as the changes in the number of plaques (Figure 2D) or fluorescent spots (Figure 2E) in the entire field of view among different experimental groups, which is a good complement to the numerical results shown in the histogram (Figure 2A and 2C).

Comments 4-9. - Figure 3: In the figure legend, the authors address a Figure 3 C. Where is this figure?

Response 4-9. I am sorry, there is no Figure 3C. The relevant description has been deleted in the revised manuscript.

Comments 4-10. - line 328: “We found that VV116 has substantial antiviral activity against the above viruses,…”. “Above viruses” is no scientific term and must be specified.

Response 4-10. The “Above viruses” refers to the various coronaviruses mentioned in the previous sentence. It is now replaced by “these coronaviruses” in the revised manuscript according to your suggestion (line 370).

Comments 4-11. - line 371-379: this section has a different font size

Response 4-11. Thank you for your reminder, it has been corrected in the revised manuscript.

Comments 4-12. The examples listed above were just a handful of many other such errors by the authors. Thus, the article should be completely revised by a native speaker.

Response 4-12. Thank you for your reminder. In order to further improve the readability of the entire article, our revised version of the manuscript have been polished by MDPI editing services.

Comments 5. Figure 3 B: In this figure the authors only show one Western blot in which no differences can be seen, especially in the low concentration ranges of VV116. In order to obtain statistical significance regarding these experiments, the authors must carry out a densitometric evaluation of all Western blots performed; otherwise, this figure has no significance whatsoever.

Response 5. Thank you for your reminder. Each western blot was performed in at least three replicates, and we have included the densitometric analysis histograms in the results based on your suggestion.

Comments 6. The results of Figure 1 show that the EC50 values of VV116 are in some cases 1-2 log stages higher than these from Remdesivir. However, the authors do not mention or even discuss this discrepancy. Do they have an explanation for this observation? They should also compare their results with the published results for the EC50 of VV116 and Remdesivir regarding SARS-CoV-2.

Response 6. Thank you for your reminder. The original intention of this research is to study the inhibitory effect and possible mechanisms of VV116 on various coronaviruses other than SARS-CoV-2. Remdesivir is a well-known antiviral nucleoside analog, and it was used as a positive control in this research, so this study did not involve a comparison of the antiviral effects between these two compounds.

According to the report by Xie et al. [7], the anti-SARS-CoV-2 EC50 of VV116 was 0.35 μM compared to 1.71 μM for Remdesivir in Vero E6 cells. In this result, the EC50 value of Remdesivir is greater than that of VV116, which is inconsistent with the trend in our results on other coronaviruses. We did not repeated the anti-SARS-CoV-2 experiments in Vero E6 cells (First, this is not our purpose in this study; second, this experiment should be conducted in a BSL-3 level facility), so we cannot directly compare this result with ours. We believe that the EC50 value of a compound will be affected by many factors. The antiviral effect of the same compound will be significantly affected when targeting different viruses, or when targeting the same virus in different host cell environments.

Comments 7. The authors compare VV116 and Remdesivir in their study. However, they never mention the chemical differences of these two compounds. It would also be very helpful for the reader when they depict a chemical structure of these two compounds.

Response 7. Thank you for your suggestion. We have added the structure diagrams and related descriptions of Remdesivir and VV116 to the manuscript (Figure 1A and line 92-97).

Comments 8. Figure 1 A need revision. The pictures and especially the lettering are too small and the reader cannot recognize anything.

Response 8. I am sorry to cause trouble for your reading. We have increased the size of all fonts in Figure 1A.

Comments 9. Line 225-227: “The explanation could be that viral protein expression occurs downstream of viral RNA expression and that the difference in viral quantity among different experimental groups is further amplified as it goes downstream.” Speculations must be shifted into the discussion section.

Response 9. Thank you for your suggestions. This text has been moved to the discussion section (line 373-377).

Comments 10. line 371: “Collectively, we demonstrated that VV116, as a nucleoside analog targeting the coronavirus RdRp,…”. This sentence must be removed, as the authors did not show in this manuscript, that VV116 directly targets and inhibits the RdRP.

Response 10. Thank you for your reminder. We did not directly demonstrate the inhibitory effect of VV116 on HCoV-NL63 RdRp in this study. This sentence only means that VV116 is a nucleoside analog, it has been proved to inhibit the RdRp of SARS-CoV-2 by others, and it is proved to have broad-spectrum anti-coronavirus activity in this study.

In order to avoid ambiguity among readers, we changed this sentence to “In summary, we demonstrated that VV116, as a nucleoside analog proven to inhibit the RdRp of SARS-CoV-2 by others, exhibited broad-spectrum anti-coronavirus activity in this study,...” (line415-416).

Once again, we sincerely thank you for your valuable comments. We benefited a lot.

Reference

  1. Khan, S.; El Morabet, R.; Khan, R.A.; Bindajam, A.; Alqadhi, S.; Alsubih, M.; Khan, N.A. Where We Missed? Middle East Respiratory Syndrome (MERS-CoV) Epidemiology in Saudi Arabia; 2012–2019. Sci Total Environ 2020, 747, 141369, doi:10.1016/j.scitotenv.2020.141369.
  2. Gao, Y.-Y.; Wang, Q.; Liang, X.-Y.; Zhang, S.; Bao, D.; Zhao, H.; Li, S.-B.; Wang, K.; Hu, G.-X.; Gao, F.-S. An Updated Review of Feline Coronavirus: Mind the Two Biotypes. Virus Research 2023, 326, 199059, doi:10.1016/j.virusres.2023.199059.
  3. Tekes, G.; Thiel, H.-J. Chapter Six - Feline Coronaviruses: Pathogenesis of Feline Infectious Peritonitis. In Advances in Virus Research; Ziebuhr, J., Ed.; Coronaviruses; Academic Press, 2016; Vol. 96, pp. 193–218.
  4. Pedersen, N.C. A Review of Feline Infectious Peritonitis Virus Infection: 1963–2008. Journal of Feline Medicine and Surgery 2009, 11, 225–258, doi:10.1016/j.jfms.2008.09.008.
  5. Coilly, A.; Roche, B.; Dumortier, J.; Leroy, V.; Botta-Fridlund, D.; Radenne, S.; Pageaux, G.-P.; Si-Ahmed, S.-N.; Guillaud, O.; Antonini, T.M.; et al. Safety and Efficacy of Protease Inhibitors to Treat Hepatitis C after Liver Transplantation: A Multicenter Experience. Journal of Hepatology 2014, 60, 78–86, doi:10.1016/j.jhep.2013.08.018.
  6. Zhang, R.; Zhang, Y.; Zheng, W.; Shang, W.; Wu, Y.; Li, N.; Xiong, J.; Jiang, H.; Shen, J.; Xiao, G.; et al. Oral Remdesivir Derivative VV116 Is a Potent Inhibitor of Respiratory Syncytial Virus with Efficacy in Mouse Model. Signal Transduct Target Ther 2022, 7, 123, doi:10.1038/s41392-022-00963-7.
  7. Xie, Y.; Yin, W.; Zhang, Y.; Shang, W.; Wang, Z.; Luan, X.; Tian, G.; Aisa, H.A.; Xu, Y.; Xiao, G.; et al. Design and Development of an Oral Remdesivir Derivative VV116 against SARS-CoV-2. Cell Res 2021, 31, 1212–1214, doi:10.1038/s41422-021-00570-1.

Round 2

Reviewer 1 Report

Comments and Suggestions for Authors

The Article presents the flaws especially about the real "novelty" and the real potential of VV116 molecule to be confirmed as a real "potent broad-spectrum inhibitor". This new version of the manuscript is real better then the previous one. 

The Authors answered to the suggestions and revisions proposed and in all sections of the manuscript there are corrections now.

Anyway, Authors do not try to change the Y-axis of Figure 4A (Comment 10-1). You can report the Figure with a Log, but if use the decimal values even at 25 mg/kg the differences are dramatic respect to the control. This is a suggestion to find a new trend of copies/gram tissue vs VV116 mg/kg and to plan some different experiments.

Comments on the Quality of English Language

The Article is improved also for the language and the explanations

Author Response

We sincerely thank you for your constructive suggestion and the detailed explanation for Comment 10-1. We will pay attention to this in subsequent VV116 studies and other pharmaceutical research.

We have benefited a lot.

Reviewer 2 Report

Comments and Suggestions for Authors

Authors have made substantial changes and have resolved all the queries. The quality of the manuscript is improved significantly. 

Author Response

We are sincerely thankful for your previous constructive suggestions, which will be of great help to our future research and article writing.

We have benefited a lot.

Reviewer 3 Report

Comments and Suggestions for Authors

my point of view the manuscript is now acceptable for publication

Comments on the Quality of English Language

.

Author Response

(The authors gave the same response as above.)
